Predation experiments with 3D-printed lizard models yield limited responses in pheasants

Smolinský Radovan radovan.smolinsky@gmail.com 1
Adam Ivo 1
Hiadlovská Zuzana 2
Sau Shubhra 3
Škrabánek Pavel 4
Martínková Natália 5 6
1 Department of Biology, Masaryk University , Brno , Czech Republic
2 Institute of Animal Physiology and Genetics, Czech Academy of Sciences , Brno , Czech Republic
3 Department of Botany and Zoology, Masaryk University , Brno , Czech Republic
4 Institute of Automation and Computer Science, Brno University of Technology , Brno , Czech Republic
5 Institute of Vertebrate Biology, Czech Academy of Sciences , Brno , Czech Republic
6 RECETOX, Masaryk University , Brno , Czech Republic
Brygadyrenko Viktor
Electronic publication date: 2025 Nov 3
Publication date: 2025
Volume: 13
Electronic Location ID: e20103
Received 2025 Jun 20; Accepted 2025 Aug 27
Copyright: ©2025 Smolinský et al.
Copyright year: 2025
Copyright holder: Smolinský et al.
License: This is an open access article distributed under the terms of the Creative Commons Attribution License, which permits unrestricted use, distribution, reproduction and adaptation in any medium and for any purpose provided that it is properly attributed. For attribution, the original author(s), title, publication source (PeerJ) and either DOI or URL of the article must be cited.
License URL: https://creativecommons.org/licenses/by/4.0/

Keywords: Predator-prey dynamics, Predation experiment, Hyperspectral imaging, Colour polymorphism, Replacement, Reduction, Refinement, 3R

Funding: Institute of Vertebrate Biology RVO:68081766 Ministry of Education, Youth and Sports LM2023069 This study was supported by the institutional funding of the Institute of Vertebrate Biology (RVO:68081766). The RECETOX Research Infrastructure (No LM2023069) financed by the Ministry of Education, Youth and Sports provided supportive background. The funders had no role in study design, data collection and analysis, decision to publish, or preparation of the manuscript.

==============================
Animal colouration has been viewed as an adaptation shaped by both abiotic and biotic factors, balancing sexual attractiveness against predation risk. In studying predator-prey dynamics, using 3D models as prey surrogates is common, but material constraints can affect outcomes in both natural and seminatural settings. Here, we utilized 3D-printed models representing three colour morphs of sand lizards (Lacerta agilis) to investigate interactions with captive-bred pheasants (Phasianus colchicus) utilizing forced exploration experiments in an outdoor arena fitted with a grass carpet. The models adequately represented the lizard colouration across a reflectance spectral range of 330–800 nm. Our findings indicate that the pheasants generally exhibited a minimal response to static models, with significant differences observed only in younger birds (7–12 weeks old), demonstrating a higher alert response than adults. No effects were found relating to the colour morph or sex of the lizard models. These results suggest that immobile 3D-printed prey models may be insufficient to trigger natural predator responses in this predator-prey system, highlighting potential limitations of static models in eliciting predator reaction.

Introduction

The evolution of animal colour patterns reflects a balance between avoiding predation and facilitating communication. Across taxa, colour traits are shaped by the need to remain hidden from predators while also signalling fitness, identity, or warning information to others (Endler, 1978; Stevens & Merilaita, 2011). This evolutionary compromise is particularly evident in environments where predator perception varies, and selection on visual signals interacts with habitat complexity, predator behaviour, and prey mobility (Stuart-Fox et al., 2003; Kjernsmo & Merilaita, 2012; Smolinský et al., 2022). This interplay between predators and their prey further drives the evolution of phenotypic traits, particularly in colouration (Abrams, 2000; Brodie III & Brodie Jr., 1999). Predators’ visual capabilities, such as the tetrachromatic vision of birds, significantly influence how prey species adapt their colour patterns to balance camouflage and conspicuousness (Pianka & Vitt, 2003; Stuart-Fox et al., 2003). This balance is especially critical during the breeding season when male nuptial colouration may increase visibility to predators while also serving as a signal in sexual selection (Husak et al., 2006).

Lizards are frequently preyed upon by raptors and omnivorous birds, including pheasants (Blanke & Fearnley, 2015; Kabisch & Belter, 1968; Elbing, Günther & Rahmel, 1996; Jedrzejewska & Jedrzejewski, 1998; Pfeifer, 1998; Poulin et al., 2001). The common pheasant (Phasianus colchicus) is an omnivorous galliform bird endemic to Asia that has been introduced as a game species. The introduction of pheasants has raised concerns due to their detrimental impact on smaller or isolated reptile populations, with potential consequences for local extinctions (Graitson & Taymans, 2022). Quantifying this impact remains challenging, as traditional methods for analyzing predation, such as direct observation or DNA analysis of faeces, have significant limitations. For example, the rapid degradation of reptile DNA during digestion complicates the identification of sand lizard (Lacerta agilis) predation by P. colchicus (Dimond et al., 2014; Regnault, Lucas & Fumagalli, 2006).

Our previous research revealed disparities in seasonal sex-specific survival and tail autotomy rates in sand lizards (L. agilis). During the spring mating and egg-laying season, females exhibited lower survival and fewer tail autotomy events, indicating that they were less likely to escape once attacked (Smolinský et al., 2022). This suggests that gravid females are more vulnerable to predation, not due to increased exposure, but because their altered body shape and reduced mobility limit their ability to flee (Shine, 1980; Magnhagen, 1991; Sinervo, Hedges & Adolph, 1991; Shine, 2003). Predation pressure from pheasants may be particularly high during this period, as endothermic predators remain effective in cooler spring conditions, while gravid lizards experience reduced locomotor performance. These seasonal and physiological factors together likely contribute to elevated predator-driven mortality in reproductive females, but ethical restrictions prevent direct testing with live prey, necessitating alternative experimental approaches. Additionally, the sexual, seasonal, and colour morph variation in sand lizard colouration complicates efforts to experimentally isolate the visual cues that influence predator detection.

Experimental simulations provide a valuable approach to studying predator–prey dynamics, particularly for species like sand lizards. Their outcomes are often comparable to trials involving live animals (Karpestam, Merilaita & Forsman, 2013; de Alcantara Viana et al., 2022). Models made from plasticine, clay or silicone, or modified toys that mimic potential lizard prey, have been instrumental in examining predator responses to various colourations (Bateman, Fleming & Wolfe, 2017; Purger et al., 2025; Rößler, Pröhl & Lötters, 2018). These studies often focus on the nuptial colouration of male lizards, cryptic colouration of females, or the conspicuous colouration of juveniles (Castilla & Labra, 1998; Marshall, Philpot & Stevens, 2015; Watson et al., 2012; Wuthrich, Nagel & Swierk, 2022).

The adoption of 3D-printed models marks a substantial advance in the study of evolutionary dynamics between predators and prey (Walker & Humphries, 2019). These models allow for precise replication of lizard spectral properties in both visible and UV spectra while eliminating potential scent confounders (Behm et al., 2018). By refining experimental tools, 3D-printed models enable researchers to rigorously investigate predator–prey interactions while adhering to the principles of the 3Rs (Replacement, Reduction, and Refinement), thus balancing scientific inquiry with ethical considerations.

Our study focuses on the interactions between pheasants and sand lizard models under controlled conditions. By isolating the effect of colouration and using 3D-printed models, we aimed to assess how different lizard colour morphs influence predator responses. We hypothesised that the visibility of certain colour morphs increases predation risk from avian predators with tetrachromatic vision. Although we aimed to assess the role of colour morphs in predator–prey dynamics, our results underscore methodological constraints in using immobile models.

Methods

3D lizard model

We used a 3D gecko template (https://www.ameede.net/gecko-figurine-b008912-file-obj-free-download-3d-model-for-cnc-and-3d-printer/), adjusted to the size and shape of Lacerta agilis in the PrusaSlicer 2.4.2 software as a model for the experiment. The final 3D model corresponded in length of 12 cm and the body shape to an adult sand lizard. The models were colour-matched to specific individuals captured in the wild, representing colour morphs typica, concolor, and erythronotus of both sexes (Fig. S1). The colour morph typica has three light-brown craniocaudal stripes on dorsal side, where the medial line is dashed, and dark spots with light centers (oceli) between the craniocaudal stripes. Oceli form 1–2 rows on lateral sides. Lateral sides are brown in females and green in males in nuptial colouration (Sau, Smolinský & Martínková, 2023). The concolor morph lacks oceli, with a uniform dorsal and lateral colouration that may have faint stripes or small darker spots; males are typically green, occasionally gray, while females are gray or sometimes greenish-brown. The erythronotus morph has a red, brown, or orange dorsal side without patterns or green pigmentation, while the lateral sides are identical to those of the typica morph (Sau, Smolinský & Martínková, 2023).

Additional three models served as controls in highly contrasting colouration (Fig. S1). We used blue, red and yellow controls because these colours are not part of the natural colouration of sand lizards and thus served as neutral stimuli. Their purpose was not to mimic food, predators, or conspecifics, but to provide a baseline for assessing whether the experimental models elicited stronger behavioural responses. Prior studies have shown that when bright colours are not associated with known threats or rewards, they can help isolate the effect of the specific experimental manipulation, such as naturalistic shape or colouration (e.g., Kivelä et al., 2023; Hernández-Agüero et al., 2020). Testing three different bright colours allowed us to minimise the risk that any single colour would incidentally bias the results due to unknown individual preferences or aversions.

We used Revell acrylic colours, sprayed with an airbrush technique, and sealed with Mr. Hobby’s varnish for painting. According to the manufacturer, Revell acrylics are water-based and non-toxic, with good surface coverage and long-term stability. These paints are commonly used in behavioural studies involving birds (Antonová, Veselý & Fuchs, 2021; Wiebe & Slagsvold, 2009), suggesting that their scent or chemical composition does not interfere with predator responses in similar experimental contexts.

3D model congruence

To evaluate the perception differences in colour between the lizard models and six live animals of the colour morph typica (three males and three females), we measured the absolute reflectance of both with a line scan hyperspectral camera Resonon PIKA NUV-503 (Resonon, Bozeman, MT, USA). The camera has a spectral range of 330–800 nm. Each lizard model or animal was placed on an 18% photographic grey card and illuminated with both visible and ultraviolet light sources. We conducted scans from a dorsal view, simulating the perspective of an approaching pheasant on two separate days. On each day, before taking measurements, we performed a dark correction to eliminate the dark noise generated by the image sensor.

Predation experiment

The predation experiments were conducted with pheasants (P. colchicus) from private breeders (for details on animal numbers, see Table S1). The animals were kept in an aviary breeding facility in Ivančice (South Moravia Region, Czechia). Two age categories were selected: adult pheasants (aged 1.5 years) and juveniles (aged 7–12 weeks). The adults were kept in separate aviaries of 6 × 3 × 2.5 m (length × width × height) in a ratio of one cock per 4 to 10 hens. Juveniles lived together in mixed-sex groups of 250 chicks in rectangular aviary enclosures of 40 × 20 m (in accordance to §8, Czech Act No. 208/2004 Coll.). The adult pheasants were fed a mixture of N1 food (ADW Agro, Krahulov, Czechia), wheat and occasionally chopped grass. Young pheasants were fed a mixture of BŽ1, BŽ2 food (ADW Agro) and wheat. Water and feed were available for both ad libitum. Pheasants were preventively treated against coccidiosis, syngamosis, and intestinal parasitoses when they arrived at the facility and then at regular intervals once per month.

To identify individual pheasants, each tested animal had a 12 mm diameter plastic lock marker ring placed on its leg. Several rings of different colours were chosen to facilitate the differentiation of individuals. In addition to colour, each ring had a unique numerical code, so no animal was used repeatedly in one day. In total, we used 91 adult and 150 juvenile pheasants in the experiments (Table S1).

The experimental arena was a cage made from chicken mesh with a size of 2.5 × 3 × 2.5 m. All sides were covered with green gardening mesh, and we used natural grass carpets as a substrate.

The experiment started at different times but always during daylight (series of trials starting in summer season from 06:30, ending not later than 20:30) by placing the lizard model into the arena in a random location (but never in the 5 cm strip along the cage wall). The models had an identification number on the underside and the order of their use in consecutive trials for the day was also randomised. After placing the model, the same person recovered a pheasant from the housing aviary and recorded its ring colour and number, sex, and age. Then, the trial started by placing the bird into the experimental arena, and the timer was set for 10 min with behaviour recorded by the camera (Samsung SMX-F50BN) mounted on a tripod outside of a corner of the experimental arena.

Preliminary experiments (Adam, 2022) showed that habituation up to one hour did not improve pheasant interaction with the 3D lizard model. Therefore, we used the forced exploration technique (Kendall, 1974) where the pheasant was gently but simply released into the arena. During trials, an observer was present and took additional notes on the details of the pheasant’s behaviour and any environmental disturbances. After the 10-minute time limit, the recording stopped, and the same person returned the pheasant back into the housing aviary. If the pheasant attacked the lizard model, the trials ended before the time limit. The arena was cleaned between the trials, the shade tarp was adjusted, and the 3D model was replaced with a new model. The same pheasant was not used repeatedly on the given experimental day. For details on the experimental schedule, see Table S1. Data could not be recorded blindly because our study involved focal animals in the field.

The video recordings were analysed by using the R package behav 0.4.1 (available at https://github.com/nmartinkova/behav). We recorded the first instances of the alert and attack behaviours during the entire 10min duration of the trial (or less in case of attack, see above). A pheasant was considered to have alerted to the model if it visually oriented toward the 3D lizard model (Video S1), and to have attacked it if it pecked the model (Video S2).

Statistical analyses

To assess congruence in colour between the lizard models and live animals, we first converted the hyperspectral images from absolute reflectance to relative reflectance. To eliminate differences in illumination between the data from the first and second day, we used the first day as the reference and calculated a gain of signal for the second day. Consequently, we multiplied the spectral profile of each pixel in each second-day image by this gain.

We determined the gain with the help of the 18% grey card. Specifically, we used small patch of each image capturing the card to calculate the average spectral profile of the card in the image. We sorted the profiles according to the days the images were captured and calculated the average spectral profiles of the card for both the first and second days. To obtain the gain of the signal, we performed an element-wise division of the average spectral profile of the first day by that of the second day and then averaged the resulting values.

We manually segmented the images to include into the analysis only pixels of the model or the sand lizard, respectively, and smoothed the spectral profile for each pixel using a Gaussian filter with a kernel length of nine samples and a standard deviation of 1.6. We used the anonymised pixel-wise spectra to cluster the colours with iterative expectation maximisation (EM) algorithm. The EM algorithm fits a Gaussian mixture model (GMM) to data assuming that the data were drawn from multiple Gaussian distributions with unknown mean and standard deviation. We fitted the GMM with the full covariance matrix structure and regularization parameter value equal to 0.01 for 3–15 clusters K. The fitting process was controlled using the log-likelihood function, with a termination tolerance of 10−6 and a maximum of 1,000 iterations. Using the elbow method, we selected the optimal number of clusters based on the Akaike Information Criteria (AIC). To evaluate whether the statistically different colours in the distinct clusters are also perceptually different for the pheasants and lizards, respectively, we transformed the mean spectral profiles in the clusters into reptile and avian visual models. This allowed us to compare the receiver perception of the visual signals.

The avian visual model was the average avian model implemented in the pavo package (avg.uv;  Maia et al., 2019) modified for the wavelength range measured in this study. We created the reptile visual model from the work of Martin et al. (2015), who measured sensitivity spectra for two lacertid species, Podarcis muralis and Zootoca vivipara. We used peak cone sensitivities of 362, 447, 492, and 586 nm, corresponding to ultraviolet, short-wave, medium-wave, and long-wave channels, respectively, and integrated the area under each sensitivity curve to 1.

To calculate the noise-weighted Euclidean distances between colours, we assumed photoreceptor densities for the cones to be 1:2:4:3 for the avian model and 1:2:6:6 for the reptile model (Hart, 2001; Martin et al., 2015). The units of the colour distances are in just noticeable differences (JND), and we assumed that the threshold for discrimination (the animals would be able to distinguish the colours as distinct) was equal to 1.

To analyze pheasant behavioural responses to the lizard models, we first calculated the latency (time from the start of the trial to the first occurrence of the respective event) for alert and attack behaviours. These metrics were extracted from scored video recordings using the behav package. No animals or lizard models were excluded during the experiment. All data points were included in the analysis, and no inclusion or exclusion criteria were established a priori. The data were right-censored for Kaplan–Meier survival analysis and Cox proportional hazards analysis.

The Kaplan–Meier estimator is a non-parametric method used to estimate the probability of survival over time, where survival refers to the probability of remaining in a particular state without experiencing a specified event. In this study, survival means the pheasant did not notice or attack the lizard model. The Kaplan–Meier curves illustrate how the probability of lizard model survival changes over time, conditioned on the pheasant’s alert or attack behaviours during the experiment. To account for potential differences in response patterns, the data were stratified by pheasant age class, as juvenile and adult pheasants may exhibit different behavioural tendencies. These curves were generated considering various predictors, including model colour morphs, model sex, pheasant age class, and disturbances on the day before the experiment, such as storms, heavy rain, the arrival of new birds, and maintenance activities within the enclosures.

Initially, we assessed the pheasant response to control models of different colours to confirm that these responses were consistent enough to be pooled for subsequent analyses. Following this, we evaluated how pheasants reacted to lizard models representing different sexes. Given that sexual dimorphism in sand lizards manifests as colour variations on the lateral sides, but our study focused on the responses to colour patterns on the dorsal side, we examined the impact of hues of green on male models and brown on female models on pheasant responses. When the responses of pheasants did not statistically differ from the models of either sex, we evaluated the influence of the model colour morphs irrespective of the model sex.

When the Kaplan–Meier survival curves do not overlap, the proportional hazards of the predictors may differ, allowing for further evaluation using Cox proportional hazards models. The assumption of proportional hazards was considered supported across all analyses when we found no significant relationships between the Schoenfeld residuals of the explanatory variables and time. This lack of significant trends suggests that the hazard ratios for our predictors are constant over time, validating the use of these models in our study.

The pre-processing of the hyperspectral datacubes and the EM clustering was performed in Matlab. The subsequent analyses were conducted in R 4.3 (R Core Team, 2024) using packages behav 0.4.1, pavo 2.9.0 (Maia et al., 2019), survival 3.7-0 (Therneau, 2024) and vegan 2.6−6.1 (Oksanen et al., 2024).

Ethical approval declarations The living lizards of both sexes captured during other research (Smolinský et al., 2022) were held in position by qualified personnel during scanning. All sampling was performed according to the permit number JMK 42819/2021 issued by the Regional Authority of the South Moravian Region, Czechia. The experiments were approved by Ethics Committee of the Institute of Vertebrate Biology, Czech Academy of Sciences and by the Ministry of the Environment, Czechia, permit number MZP/2022/630/29. Following data collection, the animals were released at the capture site.

Results

Spectral colour composition of sand lizards and 3D models

Using GMM on anonymised pixels of sand lizard males and females of the colour morph typica and the respective 3D models, we identified six, nine, or 13 different spectral profiles as suitable candidates with the elbow method from the AIC values (Fig. 1A). Multiple colour pairs for K ∈ {9, 13} were not distinguishable for the experimental animals, especially for the reptile visual model (Table S2). We consider statistically distinct colours that do not have sufficient perceptual differences (JND <1) for the experimental animals without biological relevance for our study. Therefore, we chose K = 6 for subsequent analyses (Table S2, Fig. 2A). We calculated that all six colours are mutually distinguishable for birds and lizards when accounting for the differences in their visual models (Fig. 2B).

Figure 1 Statistically distinct colours in sand lizards and 3D models representing colour morph typica, dorsal view.

(A) The number of statistically distinct colours was chosen using the Akaike Information Criterion (AIC) plot from the Gaussian mixture models with variable number of clusters K. K = 6 was chosen, as higher number of statistically differentiated colours resulted in lack of their distinctiveness in the avian and reptile visual models (Fig. 2). (B) Heatmap of frequencies of pixel assignments to clusters of colours. The 3D models (Typica-M and Typica-F) had colour composition most similar to individuals KP6A-F and KP16B-F. Dark green colour indicates high frequency. Name suffix: -F, female colouration; -M, male nuptial colouration.

The heatmap of frequencies of the colour clusters (Table S3) showed dominance of the colour 2 (blue solid line in Fig. 2) in dorsal view of the individual animals and the 3D models of the colour morph typica (Fig. 1B). The 3D models clustered together irrespective of which sex they represented and grouped with two female individuals. Males in nuptial colouration grouped with another female.

Pheasant response to control lizard models

We evaluated pheasant alert and attack responses using control lizard models in three colours: blue, red, and yellow (Fig. S1). Kaplan–Meier estimators, derived from 75 experiments, quantified the survival probability of the models (i.e., the probability of not eliciting an alert or attack behaviour from pheasants). Across all experiments, 17 alert events and two attack events were recorded (Fig. S2). For the blue model, the survival probability decreased from 0.97 at 243s (nrisk = 29, 95% confidence interval (CI) [0.90–1.00]) to 0.83 at 479s (nrisk = 25, CI [0.70–0.98]). The red model showed a decline in survival probability from 0.96 at 85s (nrisk = 26, CI [0.89–1.00]) to 0.77 at 460s (nrisk = 21, CI [0.62–0.95]), and for the yellow model, the survival probability dropped from 0.95 at 92s (nrisk = 20, CI [0.86–1.00]) to 0.70 at 345s (nrisk = 15, CI [0.53–0.93]).

The similar trends observed in survival probability across the three control colours suggest that these colours are equivalent in terms of their ability to evade pheasant alert or attack behaviours. Consequently, these control colours were considered equivalent for subsequent analyses.

Pheasant alert and attack to the lizard colour morph models

We exposed pheasants to lizard models representing both male (hues of brown and green) and female (hues of brown) colour patterns of all colour morphs. Using Kaplan–Meier survival analysis, we observed that the the survival probability (i.e., the probability of not eliciting alert or attack behaviours) decreased similarly for models representing both sexes (Fig. S3B). Specifically, for models representing female sand lizards, the survival probability for alert decreased from 0.99 at 1s (nrisk = 197, CI [0.99–1.00]) to 0.75 at 570s (nrisk = 148, CI [0.69–0.81]). Similarly, for male models, survival probability declined from 0.99 at 23s (nrisk = 206, CI [0.99–1.00]) to 0.75 at 548s (nrisk = 152, CI [0.69–0.81]).

The declines in survival probability for attack behaviours were less pronounced. Female models showed a decrease from 0.99 at 97s (nrisk = 196, CI [0.99–1.00]) to 0.94 at 572s (nrisk = 186, CI [0.91–0.98]), while male models declined from 0.99 at 107s (nrisk = 206, CI [0.98–1.00]) to 0.91 at 510s (nrisk = 185, CI [0.87–0.95]). Given these comparable trends in survival probabilities, models representing both sexes were pooled for subsequent analyses.

Among the other predictors, Kaplan–Meier curves overlapped in all cases, except for alert behaviour conditional on the pheasant age class (Fig. S3). For adult pheasants, the survival probability for alert declined from 0.99 at 46s (nrisk = 330, CI [0.99–1.00]) to 0.82 at 570s (nrisk = 272, CI [0.78–0.86]). In juvenile pheasants, the survival probability declined more steeply, from 0.99 at 1s (nrisk = 150, CI [0.98–1.00]) to 0.60 at 530s (nrisk = 88, CI [0.52–0.68]).

The primary aim of these experiments was to assess pheasant responses to different sand lizard colour morphs. To achieve this, we first examined the Kaplan–Meier survival curves to identify potential overlaps and differences among colour morphs (Fig. S3A). To further investigate these differences, we stratified the dataset by pheasant age class. However, even after stratification, the survival curves for the different colour morphs crossed, indicating a violation of the proportional hazards assumption required for Cox proportional hazards analysis (Fig. 3).

Figure 2 Spectral reflectance of statistically distinct colours found on sand lizards colour morph typica and the respective 3D models painted with acrylic paints from the dorsal view.

The line and pie-chart colours are illustrative and do not reflect the human perception of the given reflectance spectra. (A) Centroids of the clusters spectra of anonymized pixels from the hyperspectral images. (B) Noise-weighted Euclidean distances of the cluster centroids were calculated with respect to the avian and reptile visual models, respectively. Colour pairs in the circles depict comparisons of the respective spectra shown in (A). Dotted lines indicate thresholds the animals are likely to differentiate indicating that both lizards and birds are able to perceive the colours as distinct.

Figure 3 Kaplan–Meier survival curves for alert (A) and attack (B) behaviours of the pheasants exposed to the sand lizard models, stratified by pheasant age class.

This violation of proportional hazards assumption prevented us from effectively isolating and formally testing the influence of model colour morphs on pheasant responses using Cox proportional hazards models. Furthermore, the overlapping confidence intervals among the predictors, as shown in Table 1, suggest that statistical differences in survival probabilities were challenging to detect at various time points. However, the variability in survival probabilities across time points implies that the hazard ratios were not constant but instead changed over time.

Table 1 Kaplan–Meier predictions to pheasant alert and attack of the sand lizard models.

The model was stratified with respect to the pheasant age class.

Variable	time (s)	n risk	p	95% CI	time (s)	n risk	p	95% CI	
Alert					Attack	
Pheasant age (adult):									
control	92	55	0.98	[0.95, 1.00]	338	55	0.98	[0.95, 1.00]	
	460	48	0.86	[0.77, 0.95]	
typica	46	93	0.99	[0.97, 1.00]	225	93	0.99	[0.97, 1.00]	
	570	77	0.82	[0.74, 0.90]	572	89	0.95	[0.90, 0.99]	
concolor	81	94	0.99	[0.97, 1.00]	124	94	0.99	[0.97, 1.00]	
	570	75	0.79	[0.71, 0.87]	510	89	0.94	[0.89, 0.99]	
erythronotus	94	88	0.99	[0.97, 1.00]	107	88	0.99	[0.97, 1.00]	
	420	74	0.84	[0.77, 0.92]	420	82	0.93	[0.88, 0.99]	
Pheasant age (juvenile):									
control	85	22	0.96	[0.87, 1.00]	305	22	0.95	[0.87, 1.00]	
	479	14	0.59	[0.42, 0.84]	
typica	50	42	0.98	[0.93, 1.00]	97	42	0.98	[0.93, 1.00]	
	521	23	0.54	[0.41, 0.72]	368	38	0.90	[0.82, 1.00]	
concolor	23	41	0.98	[0.93, 1.00]	107	41	0.98	[0.93, 1.00]	
	530	28	0.66	[0.53, 0.82]	208	39	0.93	[0.85, 1.00]	
erythronotus	1	44	0.98	[0.93, 1.00]	180	43	0.98	[0.93, 1.00]	
	420	26	0.60	[0.47, 0.77]	297	39	0.91	[0.82, 1.00]	

We investigated the latency of pheasant alert and attack responses with respect to the pheasant age using Cox proportional hazards models, which was the only predictor for which the Kaplan–Meier analysis indicated proportionality of hazards (Fig. S3). The model for alert responses indicated that juvenile pheasants are 2.77 times more likely to alert to the lizard models than adults (Cox proportional hazards model: z = 5.55, p < 0.001; Fig. 4A). However, the proportionality of the hazards was time-dependent, as indicated by the Schoenfeld residuals test (χ2 = 7.29, df = 1, p = 0.007; Fig. 4B). For attack responses, juvenile pheasants were 1.66 times more likely to attack the lizard models than adults. Neverthless, this difference was not statistically significant (Cox proportional hazards model: z = 1.40, p = 0.16), meaning we cannot confidently conclude that pheasant age influences attack likelihood. The Schoenfeld residuals test (χ2 = 3.27, df = 1, p = 0.07) confirmed that this result was time-independent, suggesting that the hazard ratio remained stable over the duration of the experiment.

Figure 4 Cox proportional hazards model for pheasant alert responses by pheasant age.

(A) Cox proportional hazards model showing that juvenile pheasants are significantly more likely to alert to lizard models than adults are, with a hazard ratio of 2.77 (z = 5.55, p < 0.001). (B) Schoenfeld residuals test results for the alert model, indicating decreasing hazards of pheasant alert over time (χ2 = 7.29, df = 1, p = 0.007).

Discussion

Our study investigated the interactions between pheasants and 3D lizard models under controlled conditions, aiming to understand the influence of colouration in predator–prey dynamics within the context of both avian and reptile visual systems. We used a standard experimental design based on forced exploration (Kendall, 1974). Although we carefully matched the colours and patterns of the 3D models to those of live sand lizards to ensure ecological relevance, the study design proved to be ineffective, and we recorded a low number of events, when experimental animals interacted with the models (see Table S1). Despite this, we recognise two directions of justifications for the predominantly negative result. Study design and pheasant learning.

Study design limitations

The lack of movement in our 3D models was a deliberate choice to isolate the effect of colouration from other confounding factors, such as motion. Lizards typically rely on cryptic colouration and immobility as their primary anti-predator strategies, resorting to flight or autotomy only as a last line of defense (Pianka & Vitt, 2003; Watson et al., 2012; Wuthrich, Nagel & Swierk, 2022; Smolinský et al., 2022). By keeping the models stationary, our experimental design aimed to reflect this natural behaviour and align with standard methodologies in predator–prey studies, where stationary clay or silicone models are often used to examine predator responses (Bateman, Fleming & Wolfe, 2017; Olsson, 1993; Purger et al., 2025; Farallo & Forstner, 2012).

However, this focus on immobility may also explain the limited predator responses observed in our study, particularly the low number of attacks on the models (c.f. Zvereva & Kozlov, 2023). Without the cue of movement, predators may have relied solely on visual cues, highlighting the effectiveness of cryptic colouration in reducing detection and attack risk. This result underscores the importance of immobility and camouflage in lizard anti-predator strategies but also reveals a limitation of our approach: by not incorporating motion, we may have underestimated the role of dynamic visual cues in triggering predatory behaviour in pheasants (Stevens & Merilaita, 2009).

Interestingly, the conspicuous green colour of the male sand lizard’s nuptial colouration was minimally detected in our analyses (Fig. 1 for K6, the yellow line in Fig. 2). The imaging geometry, captured from a dorsal perspective, revealed only subtle differences between male and female colouration. This finding suggests that the green nuptial colouration, predominantly located on the lateral sides of males, remains largely concealed from a dorsal viewpoint. Consequently, its visibility to avian predators viewing from above may be reduced. Our data indicate that male camouflage during the breeding season is likely not significantly compromised by their nuptial colouration when viewed by avian predators.

Learning in omnivorous birds

Our results indicate a minimal preference for attacking different colour morphs of lizard models, with no significant variation observed based on either the age of the pheasant or the sex of the lizard model. However, a notable exception was observed in alert behaviour. Juvenile pheasants were significantly more likely than adults to show an alert reaction to the lizard models (Fig. 4). This behavioural difference may indicate age-related cognitive or perceptual development, where juveniles, less experienced and more reactive to unfamiliar stimuli, are more sensitive to novel shapes or cues in the environment (Niu et al., 2022; McCabe, 2019). Such increased responsiveness could be adaptive in early life stages, enhancing predator detection before the refinement of threat discrimination via experience.

This observation suggests that pheasant predation on lizards may be largely opportunistic, and influenced by the availability of alternative food sources, with considerable variation among individuals (Blanke & Fearnley, 2015; Nordberg & Schwarzkopf, 2019; Sage et al., 2020). The higher reactivity in juveniles may not necessarily translate to predatory motivation but reflects developmental caution. As birds age, social learning, especially from maternal cues, may attenuate these generalised alert responses and focus attention on ecologically relevant stimuli (Meier et al., 2017; Santilli & Bagliacca, 2019; Whiteside, Sage & Madden, 2015).

Our findings also raise interesting questions about individual variability in predator behaviour. Consistent with the review by Blanke & Fearnley (2015), which documented taxonomic variations in pheasant predation on lizards, our results suggest that individual pheasants may exhibit different levels of interest in lizard prey. These differences could stem from individual learning experiences or inherent preferences (Aigueperse, Calandreau & Bertin, 2013; Meier et al., 2017), potentially shaped by habitat-specific conditions or prior exposure to lizards as prey (Whiteside, Sage & Madden, 2016).

Conclusions

In conclusion, while our study highlights complexities of predator–prey interactions mediated by colour and visual perception, it also calls attention to critical limitations. The low number of recorded events prevented us from rigorously testing the hypothesis that colour polymorphism influences predator pressure. Future research should focus on how learning, individual experiences, and environmental variables shape predator behaviour and the visibility and effectiveness of prey anti-predation strategies. Expanding these studies to more naturalistic settings and diverse predator–prey systems could provide deeper insights into the ecological and evolutionary dynamics at play.

Supplemental Information

Supplemental Information 1 Code

Four numbered R scripts that analyse data in supplemental tables S1-S3.

Supplemental Information 2 Dataset of pheasant (Phasianus colchicus) behaviour when presented with 3D models of sand lizard (Lacerta agilis) color morphs in forced exploration experiments

Supplemental Information 3 Dataset of reflectance spectra of dominant colours on sand lizards and the respective 3D models

(K6) Six distinct colours. (K9) Nine statistically distinct colours, with multiple pairs indistinguishable under avian and reptile visual models. (K13) Thirteen statistically distinct colours, with multiple pairs indistinguishable under avian and reptile visual models.

Supplemental Information 4 Dataset of pixel assignment to the dominant colours in sand lizard and respective 3D model hyperspectral images

Supplemental Information 5 3D-printed models of sand lizards (Lacerta agilis) used in forced exploration experiments with pheasants (Phasianus colchicus)

Supplemental Information 6 Kaplan-Meier survival curves for alert (A) and attack (B) behaviors of the pheasants exposed to the control lizard models shows no relationship with the color of the control model

Supplemental Information 7 Kaplan-Meier survival curves for alert and attack behaviors of the pheasants exposed to the sand lizard models based on (A) model color morph, (B) model sex, (C) pheasant age class, and (D) disturbance on the day prior to the experiment

Supplemental Information 8 ARRIVE 2.0 Checklist

Supplemental Information 9 Illustrative video of the pheasant (Phasianus colchicus) alert behaviour towards a lizard 3D model

Supplemental Information 10 Illustrative video of the pheasant (Phasianus colchicus) attack behaviour towards a lizard 3D model

We thank Ondřej Májek and Tomáš Pavlík for valuable discussions. We used artificial intelligence service ChatGPT to improve grammar and style of the manuscript.

Additional Information and Declarations

Competing Interests

Author Contributions

Animal Ethics

Field Study Permissions

Data Availability

The authors declare there are no competing interests.

Radovan Smolinský conceived and designed the experiments, performed the experiments, authored or reviewed drafts of the article, and approved the final draft.

Ivo Adam conceived and designed the experiments, performed the experiments, authored or reviewed drafts of the article, and approved the final draft.

Zuzana Hiadlovská conceived and designed the experiments, authored or reviewed drafts of the article, and approved the final draft.

Shubhra Sau analyzed the data, prepared figures and/or tables, authored or reviewed drafts of the article, and approved the final draft.

Pavel Škrabánek conceived and designed the experiments, analyzed the data, authored or reviewed drafts of the article, and approved the final draft.

Natália Martínková conceived and designed the experiments, analyzed the data, prepared figures and/or tables, authored or reviewed drafts of the article, and approved the final draft.

The following information was supplied relating to ethical approvals (i.e., approving body and any reference numbers):

Ethics Committee of the Institute of Vertebrate Biology, Czech Academy of Sciences and by the Ministry of the Environment, Czechia.

The following information was supplied relating to field study approvals (i.e., approving body and any reference numbers):

Regional Authority of the South Moravian Region, Czechia.

The following information was supplied regarding data availability:

The dataset is available in the Supplementary Files.

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
