# Peer review of "Predation experiments with 3D-printed lizard models yield limited responses in pheasants"

_PeerJ, doi:10.7717/peerj.20103_

## Round 0.1 · original submission · Major Revisions

· Academic Editor

Major Revisions

Dear Dr. Smolinský, I ask you to carefully answer all the fundamental comments and supplement the manuscript with the necessary information. I hope that the new version of this article will be approved by the reviewers for publication.

Reviewer 1 ·

Basic reporting

no comment, see additional comments

Experimental design

no comment, see additional comments

Validity of the findings

no comment, see additional comments

Additional comments

The authors present a very interesting study exploring the use of 3D models of sand lizards and presenting them to pheasants to understand whether male and female sand lizards underly different selective pressure (due to bright lateral nuptial coloration in males). At least towards the end of the manuscript this becomes a central idea. I highly appreciate the authors thoroughness of statistical analyses. Great care is being taken in measuring and matching coloration of real lizards and the 3D models.
General:
My main concern stems from the fact that after reading this manuscript I am not sure what the original question of the study was. While the title reads like a method paper, parts of the manuscript have a much more ecology-driven core question. I think the manuscript would be much stronger with a clearer framing. I do think the title of the study sounds like a far too broad conclusion. Yes, the models did not prove effective in this particular study, but this, as the authors say themselves in the discussion, is more likely due to the specific predator-prey system chosen. The study is packed with technical information, some of which may be better suited to be in the supplemental materials. Overall, I would encourage the authors to add a video example to the manuscript as well as provide clearer behavioral definitions of the behaviors that were scored. Lastly, I think a rationale is missing for the control 3D prints.
Specific Comments:
L. 31: Conclusion is very broad and while I agree very much with the limitation of these studies, I do think that it strongly depends on the system and the addressed question and the particular predator-prey system under investigation.
L. 64: this seems to refer to 3D prints, but does this still apply given that you painted and varnished your models? Particularly, Revell colors often have a heavy scent and are not necessarily non-toxic (to be fair, most acrylic paints are not).
L 71-73: not sure is this a method paper or more about the dynamics
L. 97: I am not sure I understand what “over two days” refers to here. Maybe you could explain the rationale between taking pictures on two consecutive days.
L. 136-137: I think it would be great to upload one or two example videos, if possible, to give the reader a better understanding of the setup
L. 125: habituation
L. 136: Please define both alert and attack response clearly for the reader once (possibly demonstrating with the video examples). I could not find a behavioral definition in the manuscript- this seems crucial to understand your scoring.
L. 181: suggest to rephrase- lizard model not being alerted sounds a bit off
L: 227: why was blue red and yellow chosen as controls? None of them represent natural food and all of them could trigger neophobia or avoidance (in the case of red). Why test three colors and then pool them rather than choose one control color?

·

Basic reporting

This study explores how captive-bred pheasants respond to 3D-printed models of sand lizards representing three distinct colour morphs, within the broader context of colour polymorphism and predator-prey dynamics. Using a controlled outdoor arena, the authors tested whether colour morph and lizard sex influenced pheasant behaviour. Their results show limited predator response overall, with only juvenile pheasants demonstrating higher alert behaviour regardless of lizard morph. The authors conclude that immobile models may inadequately capture realistic predation pressures, pointing to key limitations in the use of static models in behavioural ecology. This is an interesting and timely contribution that touches on the methodological constraints of using artificial prey in ecological studies. The manuscript is generally well-written, and the experimental design is clear, although several clarifications and improvements are needed, particularly regarding theoretical framing, methods reporting, and interpretation of the findings. These major points are detailed below and in the other boxes, while comments and edits are given in the annotated PDF.

1 – The manuscript is well-written, and the English is clear throughout. The referenced literature is appropriate and includes key studies in the field. However, I have suggested a few additional references in the annotated PDF, particularly for specific points in the Introduction, which I believe would strengthen the manuscript.

2 – Figures and tables, including their captions, are clear and easy to interpret. The raw data are provided and are both complete and understandable.

3 - The theoretical background is heavily focused on the specific biological model (i.e., sand lizards and pheasants). I recommend that the first paragraph of the Introduction provide a broader overview of the evolution of colour traits, especially in the context of the trade-off between camouflage and conspicuousness. This topic has been widely studied across multiple taxa and ecological contexts, and beginning with a general framing would enhance the broader relevance of the study.

4 - The statement that female sand lizards show reduced survival during reproductive months could benefit from further elaboration. Are females more exposed to bird predators during this period? Please provide additional context for this point in the Introduction.

5 - Do the authors have specific predictions based on their hypotheses? What patterns were expected in the results? Clearly stating the hypotheses and associated predictions would help frame the study and better contextualise its findings.

Experimental design

The knowledge gap addressed by the study is well-defined and relevant. The Methods section is generally well-written and clear, though I have a few specific questions and suggestions:

1 – In Figure S1, please consider reordering the photos to match the sequence in which the morphs are introduced in the text (i.e., typica, concolour, and erythronotus).

2 - Please provide more information about the six live sand lizards used for reflectance measurements. What was their sex distribution? Were both males and females included, and if so, how many of each?

3 – Why was colour congruence assessed only for the typica morph? It would strengthen the study to explain why the other morphs were not similarly evaluated.

4 - Was pheasant feeding controlled prior to the experiment? If birds were fully satiated, what was their motivation to attack the lizard models? Clarifying this aspect is important for interpreting their behavioural responses.

5 - How was "alert" behaviour defined during the trials? Since this is a key behavioural metric in the study, it should be clearly and explicitly described in the Methods.

Validity of the findings

The results are clearly presented and well-illustrated, with effective use of figures and tables. The supplementary figures are also informative. I suggest incorporating Figure S1 into the main text, along with one or two photos of live sand lizards, to better illustrate the species and the experimental context. The Discussion, while somewhat brief, offers valid interpretations of the findings. However, two points warrant caution:

1 - Interpreting the low number of attacks as evidence of effective camouflage may be problematic, particularly since control models with highly conspicuous colours (e.g., blue and yellow) also received few attacks. This pattern may reflect aversive responses to uncommon colouration rather than a camouflage effect. I recommend a more cautious interpretation that considers alternative behavioural mechanisms.

2 - The observed difference in alarm behaviour between juvenile and adult pheasants is an important result and deserves further discussion. Since this was the only behavioural difference found across trials, it would be valuable to explore potential explanations, such as differences in learning or experience between age groups.

---

## Round 0.2 · accepted · Accept

· Academic Editor

Accept

Dear Dr. Smolinský, I congratulate you on the acceptance of this article for publication.

·

Basic reporting

The authors replied to all my previous questions and improved the paper, particularly by providing a broader Introduction, less centred on their predator-prey species, and clarifying their predictions.

Experimental design

The authors replied to all my questions, and I have no further comments on the study's experimental design.

Validity of the findings

The Discussion section was greatly improved after incorporating some of my suggestions and those of the other reviewer, particularly in discussing the differences the authors observed in the alarm responses of pheasants to lizard models between age groups. I have no other comments.